# A Hierarchically Structured Graphene/Ag Nanowires Paper as Thermal Interface Material

**DOI:** 10.3390/nano13050793

**Published:** 2023-02-21

**Authors:** Le Lv, Junfeng Ying, Lu Chen, Peidi Tao, Liwen Sun, Ke Yang, Li Fu, Jinhong Yu, Qingwei Yan, Wen Dai, Nan Jiang, Cheng-Te Lin

**Affiliations:** 1Key Laboratory of Marine Materials and Related Technologies, Zhejiang Key Laboratory of Marine Materials and Protective Technologies, Ningbo Institute of Materials Technology and Engineering (NIMTE), Chinese Academy of Sciences, Ningbo 315201, China; 2Center of Materials Science and Optoelectronics Engineering, University of Chinese Academy of Sciences, Beijing 100049, China; 3Key Laboratory of Novel Materials for Sensor of Zhejiang Province, College of Materials and Environmental Engineering, Hangzhou Dianzi University, Hangzhou 310018, China

**Keywords:** graphene paper, Ag nanowires, thermal interface materials

## Abstract

With the increase in heat power density in modern integrating electronics, thermal interface materials (TIM) that can efficiently fill the gaps between the heat source and heat sinks and enhance heat dissipation are urgently needed owing to their high thermal conductivity and excellent mechanical durability. Among all the emerged TIMs, graphene-based TIMs have attracted increasing attention because of the ultrahigh intrinsic thermal conductivity of graphene nanosheets. Despite extensive efforts, developing high-performance graphene-based papers with high through-plane thermal conductivity remains challenging despite their high in-plane thermal conductivity. In this study, a novel strategy for enhancing the through-plane thermal conductivity of graphene papers by in situ depositing AgNWs on graphene sheets (IGAP) was proposed, which could boost the through-plane thermal conductivity of the graphene paper up to 7.48 W m^−1^ K^−1^ under packaging conditions. In the TIM performance test under actual and simulated operating conditions, our IGAP exhibits strongly enhanced heat dissipation performance compared to the commercial thermal pads. We envision that our IGAP as a TIM has great potential for boosting the development of next-generation integrating circuit electronics.

## 1. Introduction

Development of integrated circuits toward miniaturization, higher intelligence, and high-power density would inevitably lead to serious heat dissipation problems [1,2,3]. The search for a way to prolong the service life and stabilize the performance of electronics, conducting proper thermal management strategy to rapidly dissipate heat and regulate the operating temperature of electronics is attracting increasing attention. However, in current thermal management system, various micro-gaps filled with low thermal conductivity air between the heat source and the heat sink would severely lower the heat dissipation efficiency [4,5,6]. Thus, thermal interface materials (TIMs) that could not only fill the gaps but also reveal high thermal conductivity have been proposed to address this issue [4,7,8]. Conventionally, TIMs are fabricated by introducing high thermal conductivity nanofiller, such as Al_2_O_3_, BN, into polymer matrixes [9,10,11]. Despite extensive efforts, the heat transport capacity of TIMs prepared in conventional ways is far from meeting the boomingly increasing thermal management requirement of modern electronics, whose power density has been growing exponentially [12]. Therefore, developing TIMs with significantly enhanced thermal conductivity has become the heart of the further progress of electronics.

Among all the emerged high thermal conductivity materials, graphene has become the most promising candidate for fabricating high-performance TIMs owing to their ultrahigh thermal conductivity and easy processing [13,14]. Represented by the graphene paper with ultrahigh in-plane thermal conductivity, great flexibility, and excellent processability, the graphene-based materials have been broadly commercialized as thermal management materials in portal and domestic electronics [15,16,17,18]. However, the through-plane thermal transport capability of graphene paper is extremely poor because of the high barrier for phonon transmission across a van der Waals interface between graphene sheets [19,20]. Meanwhile, some group IV 2D materials (such as silicene, germanene, and its layered structures (germanane (GeH), silicane, etc)) generally exist at extremely low through-plane thermal conductivity [21,22]. Exploiting graphene paper-based TIMs with enhanced through-plane thermal conductivity is of great significance and practical value; therefore, various attempts have been made to enhance the through-plane thermal conductivity of graphene papers and then promote this to other group IV 2D material paper. Functional modification on the surface of the graphene sheets can improve the interface relationship between graphene sheets, thereby increasing its through-plane thermal conductivity [23]. However, the gain brought by the total price of functional modification is not obvious, which is mainly due to the low thermal conductivity of the modified organic matter and the functional modification of the covalent price, which will destroy the continuous graphene structure. Besides the functionalization, compositing with high-thermal conductivity metal particles is considered to be a better solution to solve the low through-plane thermal conductivity of graphene paper. Normally, when the metal nanoparticles are located on the surface of thermal conductive frameworks, it is likely to construct scattering points, strongly hindering the rapid heat transmission in the bulk materials [24]. However, Qiu et al. [25] reported that the CNTs-based papers in situ deposited with Au nanoparticles showed enhanced interface heat conduction, which is inferring to benefit from the inspiration of the Au nanoparticles inducing the CNTs medium-frequency vocal mode, reassigning the carbon atom on the interface to activate the resonance with the metal particles. Li et al. [26] confirmed that Ag nanoparticles can induce low-frequency phonon mode excitation in graphene. Ag nanowires (AgNWs) with high length ratio can provide more continuous sound transmission pathways between graphene layers. Based on these reports, we believe that introducing the AgNWs into the graphene layer would significantly improve the thermal conductivity of graphene papers.

In this work, an in situ graphene/AgNWs hybrid paper (IGAP) with hierarchical structure was fabricated by vacuum filtration. First, the Ag+ was enriched on the surface of the graphene after dispersing graphene into the AgNO_3_ solution. Then, the AgNWs was nuclear and previously grown in the axial direction of graphene nanosheets. Compared to the original graphene paper, the thermal conductivity of IGAP increased slightly prior to compression. Interestingly, under 60% of the compression conditions, the through-plane thermal conductivity of IGAP can be further increased to 7.48 W m^−1^ K^−1^, more than double that of the pure graphene paper (GP). The comparative test results with the most advanced commercial TIM also prove IGAP’s excellent performance in cooling electronic equipment such as TIMs.

## 2. Materials and Methods

### 2.1. Materials

The rGO with the typical side size of 5 ± 0.3 μm and the thickness of 10 ± 0.3 nm was obtained by intercalation and exfoliation of graphite. The rGO was reduced by HI and thermal annealing. Polyethylene pyrirol (PVP, K29-32), ethylene glycol (EG), ethanol and AgNO_3_ were purchased from Sinopharm Chemical Reagent Co., Ltd. (Shanghai, China). All chemicals were used without further purification.

### 2.2. Preparation of in-situ rGO/AgNWs Hybrid Paper

AgNO_3_ (400 mg) and PVP (1000 mg) were dissolved in EG (100 mL), where the temperature was controlled at 60 °C, and magnetic mixing was used to ensure the complete dispersing of the PVP. The rGO (50 mg) solution in water was then subjected to a solvent exchange process using EG (50 mL). These two solutions were mixed by ultrasonic. Then, the mixture was treated at 160 °C for 4 h in a high-pressure reactor. The mixture was then filtered through a Teflon filter membrane (pore size: 0.5 μm) to obtain a thin paper. The paper was washed by ethanol and water to remove residual PVP and EG.

### 2.3. Preparation of Mixed rGO/AgNWs Paper (MGAP)

A total of 400 mg AgNO_3_, 1000 mg PVP and 100 mL EG were homogeneously mixed in a 250 mL beaker after vigorous mechanical stirring at 60 °C. The solution was transferred to a high-pressure reactor and reacted at 160 °C for 4 h, and the reactor was then cooled to ambient temperature. Finally, 50 mg rGO and 50 mL EG were added to the solution and dispersed through ultrasonication. The mixture was filtered with a Teflon filter membrane (pore size: 0.5 μm) to obtain a thin rGO/AgNWs paper. The MGAP was washed by ethanol and water to completely remove residual PVP and EG.

### 2.4. Characterizations

The morphologies of IGAP and GP were investigated using the scanning electronic microscope (SEM, Regulus 8230, Hitachi, Japan). Transmission electron microscopy (TEM, Talos F200X, Thermo Fisher Scientific Inc., USA.) and X-ray diffraction (XRD, Advance D8, Bruker, Germany) with Cu Kα radiation (λ = 1.5406 Å) were utilized to characterize the crystallinity and chemical composition of the IGAP, respectively. The compression tests were carried out on an electron omnipotence tester of universal testing machine (UTM, 5567A, Instron). The thermal conductivity of the IGAP can be calculated by the equation λ = α × Cp × ρ, in which thermal diffusivity (α) was measured using Hyper Flash laser thermal conductivity meter (LFA467, NETZSCH, Germany); specific heat capacity (Cp) was evaluated using a differential scanning calorimeter (DSC) (PYRIS Diamond™, Perkin Elmer, USA); and density (ρ) was calculated by the equation ρ = m/V. Infrared (IR) photos were captured using an infrared camera (Fluke, Ti480 Pro, Everett, DC, USA).

## 3. Results

Figure 1a shows the schematic of the fabrication procedure of IGAP. The graphene/AgNWs were obtained by in situ-deposited AgNWs onto the surface of the graphene nanosheets [27,28,29]. The hierarchical structure was created by a simple filtration process. To act as the nucleation sites and silver sources, Ag+ is rich on the surface of graphene sheets, and the EG serves as a reducing agent. PVP, as a structural orientation and stabilizer prepared by AgNWs, affected the heat transfer between graphene sheets and AgNWs. To remove residual PVP and EG, the paper was cleaned three times with alcohol and ionic water. In situ growth produced direct welding of AgNWs and graphene layers, thus greatly promoting the interface affinity. The morphologies of graphene before and after AgNWs growth are present in Figure 1b. Before the reaction starts, a large number of silver ions are adsorbed by graphene sheets. The IGAP constructed by AgNWs and graphene sheets were further characterized by XRD, and the results are shown in Figure 1c. The rGO reduced by HI and thermal annealing shows high quality. Compared with pure graphene paper (GP), diffraction peaks of AgNWs also appeared in the hybrid fillers, verifying the presence of AgNWs. In addition to the signal at 26.2° originating from the (002) plane of graphene sheets, other diffraction peaks at 38.1°, 44.3°, 64.4°, 77.4°, and 81.5° can be assigned to the (111), (200), (220), (311) and (222) crystal planes of AgNWs [30,31]. Benefiting from the simple filtration process, a large IGAP can be manufactured (Figure 1d, diameter: 280 mm).

The comparison of the microscopic morphologies of IGAP and GP is shown in Figure 2a–f. The IGAP and GP present an obvious hierarchical structure. Contrary to the smooth top surface of GP, there are more AgNWs on the top surface of IGAP. On the cross-sectional morphology, the GP stacked with graphene sheets is consistent with the conventional graphene paper. Graphene sheet stacking is fluffy, and most graphene sheets are not arranged in parallel alignment. It can be seen from the SEM images that the AgNWs, efficiently bridging the graphene layers, can provide better phonon transmission channels. AgNWs with a high aspect ratio are connected to graphene layers to promote heat transfer that is better in the through-plane direction.

In order to explore the effects of the IGAP hierarchical structure on thermal performance, the thermal diffusivity rate of IGAP, MGAP, and GP was measured through the laser flash method (α). The through-plane thermal diffusivity (TD) of the original IDAG (18 mm^2^ s^−1^) is higher than that of GP (14.8 mm^2^ s^−1^). In contrast, the thermal conductivity (TC) is close. As shown in Table 1, the corresponding thermal conductivity (κ) can be calculated by the equation κ = α × ρ × Cp, where ρ is the sample density and Cp is the specific heat. Compared to graphene, AgNWs have higher TD and lower Cp. In electronic packaging, a packing pressure was applied to ensure good contact between the mating surface and TIMs to eliminate the micro-gaps at the interface. As shown in Figure 3a, the graphene sheets in the uncompressed GP are not completely arranged at a fully horizontal level. After compression, the TD of the GP is ∽60.3% lower than the original GP, which is due to the change in graphene sheets’ orientation after pressure. This is because graphene tablets would be rearranged in the horizontal direction in the process of vertical compression. However, the reduction in TD of compressed IGAP (21.1%) is lower than that of compressed GP (60.3%). The in situ AgNWs reduce the phonon diffusion barrier across the basal planes of graphene. Interestingly, we noticed that the TC of IGAP (7.48 W m^−1^ K^−1^) after compression was ∼2.14 times higher than that of GP (3.50 W m^−1^ K^−1^). The density of IDGA was greatly increased after compression (0.49 g cm^−3^→1.18 g cm^−3^), thus compensating for the reduction in the transparent heat conduction capacity caused by the rearrangement of graphene sheets. To the best of our knowledge, IGAP has the higher through-plane thermal conductivity compared with the other graphene-based papers fabricated by filtration or in situ growth in the previous literature (in Table 2) and is also much better than that of commercial TIMs. Although covalent bond link has higher thermal conductivity, its production process is more complex, and it is difficult to realize commercial application. As presented in Figure 3c, the IGAP shows better mechanical compression performance than GP. Under 60% of the compression conditions, the compressive stress of IGAP is only 16.8 KPa, and the compressive stress of GP is 61.8 KPa, which can be explained by the following factors: the 3D network formed by the AgNWs with a high aspect ratio and graphene is fluffier than GP, and AgNWs has better bending. Lower package pressure can prevent the chip from damaging. As shown in Figure 3d, in order to compare the through-plane surface thermal transfer between IGAP and GP, pieces of 10 mm × 10 mm × 0.1 mm paper were placed on a ceramic heater (50 W) and originally kept at room temperature, followed by heating at the same time, which was recorded by calibrated infrared (IR) camera and thermocouple. The papers were sprayed with graphite on the upper surface to maintain the same infrared emission rate. The calibration of the surface temperature evolution and heating time on the surface temperature of the infrared thermal image were used. Figure 3e shows the time dependence of its surface temperature evolution on the heating time. The surface temperature of IGAP increases higher and shows a higher value than the surface temperature of the GP, which confirms IGAP has superior through-plane thermal conductivity. As shown in Figure 3f, the thermal conductivity decreases with the temperature increase. It is widely known that when the temperature rises, the electron and crystal heat motion in pure metal is intensified, and the free electron movement is blocked. As the temperature rises, the thermal conductivity of AgNWs will decrease. When the temperature is higher than 25 °C, the thermal conductivity of the graphene is negatively related to the temperature due to Umklapp phonon scattering. Figure 3g shows the thermal conductivity changes with 15 heating/cooling cycles between 25 °C and 100 °C. In the process of testing, the thermal conductivity fluctuations at the same temperature are less than 0.08 W m^−1^ K^−1^, indicating the preeminent thermal reliability and stability of IGAP.

Based on its high-plane heat conduction rate, IGAP may have the potential of high-performance TIM, which can effectively transmit heat on the heated water sink interface [37]. Therefore, as shown in Figure 4a, the verification system for the simulation heat dissipation process for electronic components is developed and compared with the most advanced thermal pads (a classic conventional TIM, approximately 5 W m^−1^ K^−1^, 5000S35, Bergquist). The IGAP and 5000S35 with a size of Φ10 mm are placed between the ceramic heater and the TGE radiator, respectively, and the thickness (BLT) of the bonding line (BLT) is 40 μm, and the vertical pressure is 20 kPa. A chilling system was used to keep the radiator at room temperature while evaluating the heater temperature. The saturation temperature of the heater and the power applied are depicted in Figure 4b; the slope value is 1.56 (without TIM), 1.23 (5000S35) and 0.99 °C cm^2^ W^−1^ (IGAP). This indicates that compared with the system without TIM and with 5000S35 thermal pads, the cooling efficiency of IGAP has increased by 45.5% and 27.3%. In Figure 4c, after the heater (25 W cm^−2^) was turned on with the start of 300 s, the temperature of the heater rose rapidly, and then reached a balance. Obviously, compared with the temperature without TIM, the cooling performance of IGAP as TIM was reduced by 42 °C at 900 s, which is much better than the cooling performance of 5000S35 thermal pads (34 °C). Then, the commercial computing fluid dynamics software (ANSYS ICEPAK) was used for testing the heat analysis of test configuration at 25 W application power (Figure 4d). The background was set to 1 atm and 25 °C in the atmosphere. The cooling medium was water with the volume flow of 250 mL min^−1^ at 25 °C. Table 3 shows the detailed settings of the heat sink and the heater, which can be estimated as effective thermal conductivity conditions (Figure 4e). Based on the effective thermal conductivity of the IGAP (~2.06 W m^−1^ K^−1^) and 5000S35 (~1.1 W m^−1^ K^−1^), not only does the IGAP have superior through-plane thermal conductivity, but also the heating contact (two sides) of the IGAP (35 K mm^2^ W^−1^) is significantly lower than the side of 5000S35 thermal pads (68 K mm^2^ W^−1^). As a result, a soft and compressible IGAP is easier to use in order to fill the micro gaps between the mating surface under the packaging, which leads to the maximum contact area and lowers thermal contact resistance at the microscope.

## 4. Conclusions

A hybrid graphene paper fabricated via filtration shows a characteristic structure in which the AgNWs connect graphene layers to enhance through-plane thermal conductivity. Ag^+^ would be enriched on the surface of the graphene to achieve in situ preparing. At low compression pressures, the through-plane thermal conductivity of IGAP reaches 7.48 W m^−1^ K^−1^, more than twice of that of the graphene paper. Therefore, we have proved that in the TIM performance test, the temperature of the heater using the compressed IGAP is reduced by 42 °C, which is superior to the 34 °C of the commercial thermal pads (5000S35, Bergquist). Under actual application conditions, IGAP, without aging like traditional polymers, has huge potential to be used as the new generation high-performance TIMs with good thermal stability. Meanwhile, inserting high TC materials into other layered materials is also of great significance to enhance the through-plane TC of composite materials.

## Figures and Tables

**Figure 1 nanomaterials-13-00793-f001:**
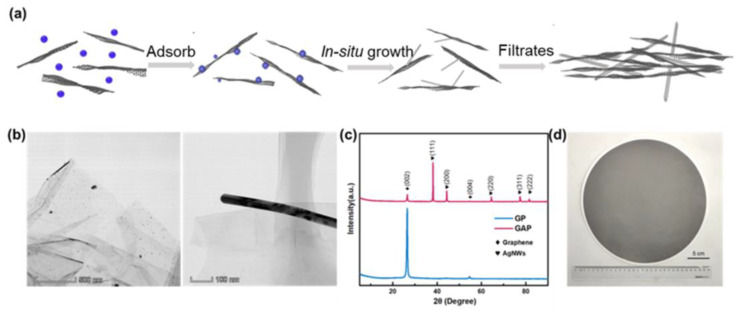
(**a**) Schematic of the fabrication process of IGAP. (**b**) The TEM images of graphene/silver ions and graphene/AgNWs. (**c**) The XRD patterns of IGAP and GP. (**d**) The photograph of IGAP.

**Figure 2 nanomaterials-13-00793-f002:**
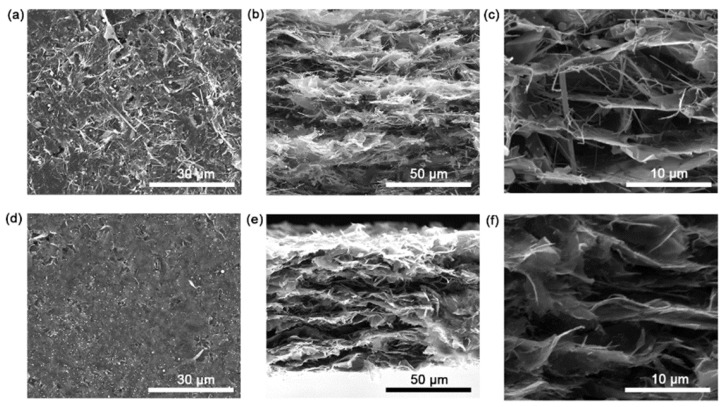
The top-view and cross-sectional SEM images of (**a**–**c**) IGAP and (**d**–**f**) GP.

**Figure 3 nanomaterials-13-00793-f003:**
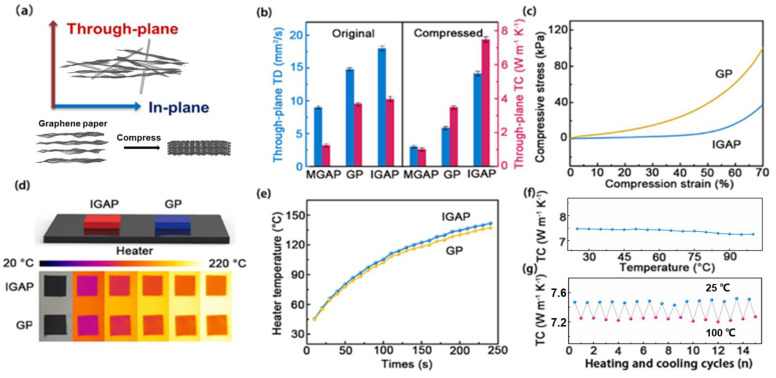
(**a**) The compression schematic of IGAP and GP. (**b**) Through-plane thermal properties of MGAP, GP and IGAP. (**c**) The stress-strain curves of IGAP and GP. (**d**) The test system configuration and corresponding IR images of IGAP and GP. (**e**) Surface temperature evolution of IGAP and GP. (**f**,**g**) The through-plane thermal conductivity as a function of environmental temperature and during the cyclic heating/cooling test.

**Figure 4 nanomaterials-13-00793-f004:**
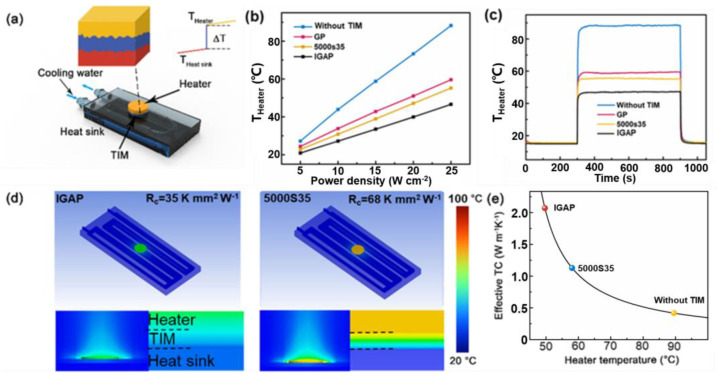
(**a**) Schematic configuration of TIM performance evaluation system. Temperature evolution of heater under various applied powers (**b**) and over time at 25 W cm^−2^ (**c**). (**d**) Comparison of heat dissipation capacity based on simulation curve. (**e**) The calculated effective thermal conductivity as a function of heater temperature.

**Table 1 nanomaterials-13-00793-t001:** The thermophysical properties of GP and IGAP.

Sample	TD mm^2^ s^−1^	Density g cm^−3^	Cp J g^−1^ K^−1^	TC W m^−1^ K^−1^
GP	14.8	0.33	0.754	3.7
	5.88	0.79		3.5
IGAP	18	0.49	0.447	3.97
	14.2	1.18		7.48

**Table 2 nanomaterials-13-00793-t002:** Comparison of thermal conductivity based on graphene papers prepared by various methods.

Method	Name	TC W m^−1^ K^−1^	Ref.
Filtration	Exfoliated graphite nanoplatelet paper Carbon nanotube–graphene hybrid paper Nanodiamond decorated functionalized graphene oxide paper Hierarchically structured graphene paper	1.3 0.2 0.3 12.6	[32] [33] [34] [19]
In-situ growth	Au NPs decorated graphene nanoplatelet paper Ag nanoparticle-intercalated graphene paper Carbon nanoring–graphene hybrid paper Graphene hybrid paper	1.6 3.3 5.8 17.6	[23] [35] [36] [20]
In-situ growth + Filtration	Graphene/Ag Nanowires paper	7.48	This work

**Table 3 nanomaterials-13-00793-t003:** The detailed settings of the heat sink and the heater.

	Materials	Size cm^3^	TC W m^−1^ K^−1^	Cp J g^−1^ K^−1^
Heat sink Heater	Aluminum alloy	Φ1*0.2	205	0.88
	Alumina	10*4.5*1.5	27	0.86

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
