# Peer review of "A Hierarchically Structured Graphene/Ag Nanowires Paper as Thermal Interface Material"

_nanomaterials, 2023, doi:10.3390/nano13050793_

Round 1

Reviewer 1 Report

The manuscript entitled ‘A Hierarchically Structured Graphene/Ag Nanowires Paper as Thermal Interface Materials’ reports Graphene/Ag Nanowires Paper as an efficient thermal interface material (TIMs). I have the following comments on this manuscript.

1)     Authors mentioned that the thermal diffusivity of the IGAP and GP decrease after compression, which is counter intuitive. They attributed it to the rearrangement of the graphene flakes in the horizontal direction in the process of vertical compression. This reasoning does not seem to be convincing as the diffusivity should increase with the rearrangement of the graphene flakes. The images of the IGAP and GP papers also show that the flakes come closer after compression, which should lead to the increased thermal diffusivity. Could authors confirm their experimental measurements or explain with a different reasoning.

2)     How did the authors calculate the thermal conductivity values of the different TIM materials, GP and IGAP? They should report the detailed calculations with the values of thermal diffusivity, specific heat capacity, and density in a table. They may present it in a supplementary file.

3)     Fig. 3e shows an increase in the values of thermal conductivity with temperature. The authors have not explained this observation. They should explain it. Will it continue to increase with temperature?

4)     In Table 1., authors have reported some graphene-based TIM materials with low thermal conductivity, which illusively makes their results interesting. There are several other examples of graphene-based TIM materials in literature exhibiting a much higher values of thermal conductivity. They should modify their table and explain their results accordingly.

5)     What is IDAG? The authors have probably mistaken with the abbreviation.

6)     Language and grammar of the manuscript need to be polished.

Author Response

Dear editor,

We are very grateful to your and the reviewers’ critical comments and thoughtful suggestions. Based on these comments and suggestions, we have made careful revision on the original manuscript entitled " A Hierarchically Structured Graphene/Ag Nanowires Paper As Thermal Interface Materials " (Manuscript ID: nanomaterials-2163052). We responded point by point to the comments of reviewer #1 as listed below, along with a clear indication of the location of the revision.

In specific response to the points raised by reviewer #1:

 Question 1: Authors mentioned that the thermal diffusivity of the IGAP and GP decrease after compression, which is counter intuitive. They attributed it to the rearrangement of the graphene flakes in the horizontal direction in the process of vertical compression. This reasoning does not seem to be convincing as the diffusivity should increase with the rearrangement of the graphene flakes. The images of the IGAP and GP papers also show that the flakes come closer after compression, which should lead to the increased thermal diffusivity. Could authors confirm their experimental measurements or explain with a different reasoning.

Reply: Thank you for the comment. Graphene is a material with anisotropic thermal conductivity (in-plane: 3500-5300 W m-1 K-1; through-plane: 1-3 W m-1 K-1). With the rearrangement of the graphene flakes, although the flakes become closer after compression, the diffusivity in though-plane direction also decreases. The increase in smaller pitch is smaller than the loss of rearrangement. The authors test the in-plane thermal diffusivity of GP, which rose from 80 mm2 s-1 to 158 mm2 s-1. The rearrangement is benefitted to the in-plane thermal diffusivity.

Question 2: How did the authors calculate the thermal conductivity values of the different TIM materials, GP and IGAP? They should report the detailed calculations with the values of thermal diffusivity, specific heat capacity, and density in a table. They may present it in a supplementary file.

Reply: Thank you for the comment. The authors revised it according to the suggestions of the reviewers. The corresponding description also have been added in the revised manuscript. (See Table 2,line 225)

Sample

TD  mm2 s-1

Density  g cm-3

Cp J g-1 K-1

TC   W m-1 K-1

GP

14.8

0.33

0.754

3.7

5.88

0.79

3.5

IGAP

18

0.49

0.447

3.97

14.2

1.18

7.48

Table 2. The thermophysical properties of GP and IGAP

Question 3: Fig. 3e shows an increase in the values of thermal conductivity with temperature. The authors have not explained this observation. They should explain it. Will it continue to increase with temperature?

Reply: Thanks for the comments. The authors re-measure the Cp, apologize for the mistake and correct it to the revised manuscript. When the temperature rises, the electron and crystal heat motion in pure metal are intensified, and the free electron movement is blocked. As the temperature rises, the thermal conductivity of AgNWs will decrease. When the temperature is higher than 25 ° C, the thermal conductivity of the graphene is negatively related to the temperature due to Umklapp phonon scattering. The thermal conductivity with IGAP will decrease with temperature rise. The corresponding description also have been added in the revised manuscript. (line 212-217)

Question 4: In Table 1, authors have reported some graphene-based TIM materials with low thermal conductivity, which illusively makes their results interesting. There are several other examples of graphene-based TIM materials in literature exhibiting a much higher values of thermal conductivity. They should modify their table and explain their results accordingly.

Reply: Thank you for the comment. The authors revised it according to the suggestions of the reviewers. The corresponding description also have been added in the revised manuscript. (See Table 1,line 222)

Table 1. Comparison of thermal conductivity based on graphene papers prepared by various methods

Method

Name

TC W m-1 K-1

Ref.

Filtration

Exfoliated graphite nanoplatelet paper

Carbon nanotube-graphene hybrid paper

Nanodiamond decorated functionalized graphene oxide paper

hierarchically structured graphene paper

1.3

0.2

0.3

12.6

[30]

[31]

[32]

[19]

In-situ growth

Au NPs decorated graphene nanoplatelet paper

1.6

3.3

5.8

17.6

[21]

[33]

[34]

[20]

Ag nanoparticle-intercalated graphene paper

Carbon nanoring-graphene hybrid paper

graphene hybrid paper

In-situ growth +

Filtration

Graphene/Ag Nanowires Paper

7.48

This work

Question 5: What is IDAG? The authors have probably mistaken with the abbreviation.

Reply: Thank you for the comment. The authors apologize for the mistake and corrected it to IGAP on the revised manuscript. The corresponding description also have been added in the revised manuscript. (line 153)

Question 6: Language and grammar of the manuscript need to be polished.

Reply: Thanks for the comments, we apologize for the language issues that appear in our manuscript. We worked on the manuscript for a long time and the repeated addition and removal of sentences and sections obviously led to poor readability. We have now corrected for spelling; grammatical and formatting issues and we really hope that the flow and language level have been substantially improved.

We appreciate for your warm work earnestly, and hope that the correction will meet with approval. The manuscript has been overall checked, and the changes marked in red. We hope that these revisions are sufficient to make our manuscript acceptable for publication in Nanomaterials. If you believe that any additional clarifications need to be addressed, I will be happy to include them. Once again, thank you very much for your comments and suggestions.

Yours sincerely,
Cheng-Te Lin
Ningbo Institute of Materials Technology and Engineering, Chinese Academy of Sciences
E-mail: [email protected]

Reviewer 2 Report

The paper is interesting and should be proceed further. Hovewer, I have some suggestions to improve it.

My comments are presented below:

The references contain only 3 items from the last two years. Literature studies should be expanded with new items.

There are group citations in the Introduction - their description should be extended.

Figure 2 should be better described in the text

The description of Figures 3 and 4 is also insufficient, and the component parts of the figure are small and hard to read

Did the authors check the mechanical strength of the mentioned hybrid graphene paper?

The authors have written in the conclusions: "Under actual operating conditions, IGAP which gets rid of the aging of traditional polymers has great potential to be used as high-performance TIMs with good thermal and chemical stability." - please provide additional explanations regarding chemical stability

Author Response

Dear editor,

We are very grateful to your and the reviewers’ critical comments and thoughtful suggestions. Based on these comments and suggestions, we have made careful revision on the original manuscript entitled " A Hierarchically Structured Graphene/Ag Nanowires Paper As Thermal Interface Materials " (Manuscript ID: nanomaterials-2163052). We responded point by point to the comments of reviewer #2 as listed below, along with a clear indication of the location of the revision.

In specific response to the points raised by reviewer #2:

 Question 1: The references contain only 3 items from the last two years. Literature studies should be expanded with new items.

Reply: Thank you for the comment. The authors revised it according to the suggestions of the reviewers. We have increased the corresponding references.

Question 2: Figure 2 should be better described in the text. The description of Figures 3 and 4 is also insufficient, and the component parts of the figure are small and hard to read.

Reply: Thank you for the comment. The authors will revise it according to the suggestions of the reviewers. The corresponding description also have been added in the revised manuscript. (line 156-220)

Question 3: Did the authors check the mechanical strength of the mentioned hybrid graphene paper?

Reply: Thank you for the comment. The TIM is mainly used to fill the gaps, so its compression performance is more important. At the same time, the existence of thermal conductive interlayers is inevitable that the mechanical properties are bound to cause adverse effects.

Question 4: The authors have written in the conclusions: "Under actual operating conditions, IGAP which gets rid of the aging of traditional polymers has great potential to be used as high-performance TIMs with good thermal and chemical stability." - please provide additional explanations regarding chemical stability.

Reply: Thank you for the comment. The authors apologize for the mistake and delete the chemical stability which needs long-term testing time. The corresponding description also have been added in the revised manuscript. (line 272-274)

We appreciate for your warm work earnestly, and hope that the correction will meet with approval. The manuscript has been overall checked, and the changes marked in red. We hope that these revisions are sufficient to make our manuscript acceptable for publication in Nanomaterials. If you believe that any additional clarifications need to be addressed, I will be happy to include them. Once again, thank you very much for your comments and suggestions.

Yours sincerely,
Cheng-Te Lin
Ningbo Institute of Materials Technology and Engineering, Chinese Academy of Sciences
E-mail: [email protected]

Reviewer 3 Report

This manuscript reports a study of AgNW graphite composite for thermal interface materials application. This language issue is seriously affecting the quality and readability of the manuscript to a level that I am almost guessing their meaning all the time. The resulting performance of the composite might be of some value for TIM use and might be considered for publication on this journal or a sister journal after major revision and language polishing.   

1 In section 2.2, what does the author mean by solvent exchange when they actually used the same solvent of EG in the second step? 

2 The claim that the AgNW bridge the graphene flakes vertically in Fig. 1(a) is not convincing based on the SEM in Fig. 2(b). It is hard to maintain the orientation of the AgNWs in reality.  

3 In line 133, what does IDAG mean? It comes out of nowhere.

4 Laser flash measurement is very sensitive to the samlpe thickness. It doesn't work when the sample is too thin. Based on the SEM, the GP seems to be only 100 um which makes me doubt the reliability of the thermal conductivity measurement results. 

5 How is the temperature of the samples surface measured in Fig. 3(d and e)? If they authors want to demonstrate that the surface temperature of IGAP is lower, the emissivity value of the GP and IGAP sample must be obtained in the range of the IR camera first since they apparently cannot use the heater stage temperature for calibration. Then what value of emissivity is used in Fig 3(e)? How can the authors assume that the IGAP has the same emissivity as GP? The  reduction of emissivity deu to the addition of AgNW may explain the higher temperature in (e) which invalidate what  the authors claimed.

6 How is the thermal conductivity vs. temperature measured in Fig 3(f)? What is the max temperature rise induced by laser flash measurement in the samples? 

7 The details of the simulation in Fig. 4 is lacking. What is the coolant flow  rate? What hs boundary condition for top surface? What are the interface thermal resistance of the IGAP/heater and IGAP/heat sink interfaces? The cartoon in Fig 4(a) makes no sense since there will be significant temperature drop across the IGAP as a thick TIM. The authors naive treated the TIM as a single interface. 

8. The English of the manuscript is far from the acceptable level of for publication with numerous grammar issues and confusing sentences. 

For example, the first 2 sentences of the introduction are both wrong. What does "avoid working" mean? 

In line 48-49, what does "total price" mean? In line 50, wha is covalent price? Do they mean covalent bonds or what? In line 51, the sentence is coimpletely nonsense. What does high-thermal meawn? High thermal conductivity? How can one use particles as a "thermal conductivity"?

In line 55, the subject of the sentence is ambiguous. In line 58, what is vocal mode? I guess the authors meant phonon or acoustic mode.  

Author Response

Dear editor,

We are very grateful to your and the reviewers’ critical comments and thoughtful suggestions. Based on these comments and suggestions, we have made careful revision on the original manuscript entitled " A Hierarchically Structured Graphene/Ag Nanowires Paper As Thermal Interface Materials " (Manuscript ID: nanomaterials-2163052). We responded point by point to the comments of reviewer #3 as listed below, along with a clear indication of the location of the revision.

In specific response to the points raised by reviewer #3:

 Question 1: In section 2.2, what does the author mean by solvent exchange when they actually used the same solvent of EG in the second step?

Reply: Thank you for the comment. The graphene was dispersed in water and the water needs to be replaced by EG.

Question 2: The claim that the AgNWs bridge the graphene flakes vertically in Fig. 1(a) is not convincing based on the SEM in Fig. 2(b). It is hard to maintain the orientation of the AgNWs in reality.

Reply: Thank you for the comment. The authors will correct Fig. 1(a) according to the suggestions of the reviewers.

Question 3: In line 133, what does IDAG mean? It comes out of nowhere.

Reply: Thank you for the comment. The authors apologize for the mistake and correct it into IGAP on the revised manuscript. (line 153)

Question 4: Laser flash measurement is very sensitive to the samlpe thickness. It doesn't work when the sample is too thin. Based on the SEM, the GP seems to be only 100 um which makes me doubt the reliability of the thermal conductivity measurement results.

Reply: Thank you for the comment. LFA 467 of Netzsch is a precision device, which technical parameters are as follows: sampling time can be as low as 1 ms and the thickness of the sample can be as thin as 10 μm.

Question 5: How is the temperature of the samples surface measured in Fig. 3(d and e)? If they authors want to demonstrate that the surface temperature of IGAP is lower, the emissivity value of the GP and IGAP sample must be obtained in the range of the IR camera first since they apparently cannot use the heater stage temperature for calibration. Then what value of emissivity is used in Fig 3(e)? How can the authors assume that the IGAP has the same emissivity as GP? The reduction of emissivity due to the addition of AgNWs may explain the higher temperature in (e) which invalidate what the authors claimed.

Reply: Thank you for the comment. In order to maintain the same infrared emission rate, a layer of graphite was sprayed on the upper surface of the two samples. The authors apologize for not mentioning it in the original manuscript and will revise it according to the suggestions of the reviewers. (line 205-206)

Question 6: How is the thermal conductivity vs. temperature measured in Fig 3(f)? What is the max temperature rise induced by laser flash measurement in the samples?

Reply: Thanks for the comments. The authors re-measure the Cp, apologize for the mistake and correct it to the revised manuscript. In laser flash measurement, the max temperature rise of the samples is depended on the thermal conductivity of sample. (line 212-217)

Question 7: The details of the simulation in Fig. 4 is lacking. What is the coolant flow rate? What has boundary condition for top surface? What are the interface thermal resistance of the IGAP/heater and IGAP/heat sink interfaces? The cartoon in Fig 4(a) makes no sense since there will be significant temperature drop across the IGAP as a thick TIM. The authors naive treated the TIM as a single interface.

Reply: Thanks for the comments, The authors apologize for not mentioning it in the original manuscript and will revise it according to the suggestions of the reviewers. (See Table 3,line 233)

Table 3. The detailed settings of the heat sink and the heater

Materials

Size  cm3

TC  W m-1 K-1

Cp  J g-1 K-1

Heat sink

Heater

Aluminum alloy

φ1*0.2

205

1

Alumina

10*4.5*1.5

27

1

Question 8: he English of the manuscript is far from the acceptable level of for publication with numerous grammar issues and confusing sentences. For example, the first 2 sentences of the introduction are both wrong. What does "avoid working" mean? In line 48-49, what does "total price" mean? In line 50, what is covalent price? Do they mean covalent bonds or what? In line 51, the sentence is completely nonsense. What does high-thermal mean? High thermal conductivity? How can one use particles as a "thermal conductivity"? In line 55, the subject of the sentence is ambiguous. In line 58, what is vocal mode? I guess the authors meant phonon or acoustic mode.

Reply: Thanks for the comments, we apologize for the language issues that appear in our manuscript. We worked on the manuscript for a long time and the repeated addition and removal of sentences and sections obviously led to poor readability. We have now corrected for spelling; grammatical and formatting issues and we really hope that the flow and language level have been substantially improved.

We appreciate for your warm work earnestly, and hope that the correction will meet with approval. The manuscript has been overall checked, and the changes marked in red. We hope that these revisions are sufficient to make our manuscript acceptable for publication in Nanomaterials. If you believe that any additional clarifications need to be addressed, I will be happy to include them. Once again, thank you very much for your comments and suggestions.

Yours sincerely,
Cheng-Te Lin
Ningbo Institute of Materials Technology and Engineering, Chinese Academy of Sciences
E-mail: [email protected]

Reviewer 4 Report

Please find attached the review of the article。

Author Response

Dear editor,

We are very grateful to your and the reviewers’ critical comments and thoughtful suggestions. Based on these comments and suggestions, we have made careful revision on the original manuscript entitled " A Hierarchically Structured Graphene/Ag Nanowires Paper As Thermal Interface Materials " (Manuscript ID: nanomaterials-2163052). We responded point by point to the comments of reviewer #4 as listed below, along with a clear indication of the location of the revision.

In specific response to the points raised by reviewer #4:

 Question 1: The English is rather bad and should be thoroughly proved.

Reply: Thanks for the comments, we apologize for the language issues that appear in our manuscript. We worked on the manuscript for a long time and the repeated addition and removal of sentences and sections obviously led to poor readability. We have now corrected for spelling; grammatical and formatting issues and we really hope that the flow and language level have been substantially improved.

Question 2: The method of graphene oxide reduction and relevant conditions should be described at last shortly.

Reply: Thank you for the comment. The rGO reduced by HI and thermal annealing shows high quality. We added it to the revised manuscript. (See Line 96-97)

Question 3: The thermal diffusion and thermal diffusivity are not the same terms, as the authors believe.

Reply: Thank you for the comment. The authors apologize for the mistake and correct thermal diffusion into thermal diffusivity on the revised manuscript. (See Line 173)

We appreciate for your warm work earnestly, and hope that the correction will meet with approval. The manuscript has been overall checked, and the changes marked in red. We hope that these revisions are sufficient to make our manuscript acceptable for publication in Nanomaterials. If you believe that any additional clarifications need to be addressed, I will be happy to include them. Once again, thank you very much for your comments and suggestions.

Yours sincerely,
Cheng-Te Lin
Ningbo Institute of Materials Technology and Engineering, Chinese Academy of Sciences
E-mail: [email protected]

Reviewer 5 Report

The authors developed the TIM with high through-plane thermal conductivity (TC) by fabricating graphene-based papers with Ag NWs. In general, graphene has large TC along the plane and low TC across the plane. The authors increase TC across the plane by using Ag NWs up to 7.48 W/mK. At first look, I think that this increase may be small, but the authors proved the effectiveness of this paper by using TIM performance evaluation system, which validation convinced me of the effectiveness.

This paper is very useful and informative in this research field. However, there are some lacks of discussion and comparison with conventional ones in the present manuscript. So, there are just two concerns as shown below.

In conclusion, the present manuscript should be revised by answering the comments. This study would meet the criteria for the publication if it is properly revised.

(1) In terms of the application, there are conventional TIMs. However, the authors compared with only GP and 5000s35. Experimental comparison may be difficult, but the discussion should be done by comparison with conventional TIMS and in introduction session, they should also be mentioned.

(2) In terms of the scientific view, the authors’ idea, namely insertion of high TC materials into gap of layered materials is main message of this paper. The universality and application to the other materials should be mentioned. For example, there are some group IV 2D materials (silicene, germanene, and its layered structures (germanane(GeH), silicane or something). Actually, thermal conductivity research of germanane was reported. The authors should discuss the universality of the idea by mentioning the application to these other group IV layered structures.

Author Response

Dear editor,

We are very grateful to your and the reviewers’ critical comments and thoughtful suggestions. Based on these comments and suggestions, we have made careful revision on the original manuscript entitled " A Hierarchically Structured Graphene/Ag Nanowires Paper As Thermal Interface Materials " (Manuscript ID: nanomaterials-2163052). We responded point by point to the comments of reviewer #5 as listed below, along with a clear indication of the location of the revision.

In specific response to the points raised by reviewer #5:

 Question 1: In terms of the application, there are conventional TIMs. However, the authors compared with only GP and 5000s35. Experimental comparison may be difficult, but the discussion should be done by comparison with conventional TIMS and in introduction session, they should also be mentioned.

Reply: Thank you for the comment. 5000S35 (Bergquist) made of a polymer mixed with high heat-conducting ceramic fillers is a classic conventional TIM, so we compared with 5000S35.

Question 2: (2) In terms of the scientific view, the authors’ idea, namely insertion of high TC materials into gap of layered materials is main message of this paper. The universality and application to the other materials should be mentioned. For example, there are some group IV 2D materials (silicene, germanene, and its layered structures (germanane (GeH), silicane or something). Actually, thermal conductivity research of germanane was reported. The authors should discuss the universality of the idea by mentioning the application to these other group IV layered structures.

Reply: Thank you for the comment. The authors have discussed the universality of the idea by mentioning the application to these other group IV layered structures on the introduction. The corresponding description also have been added in the revised manuscript. (line 58-63)

We appreciate for your warm work earnestly, and hope that the correction will meet with approval. The manuscript has been overall checked, and the changes marked in red. We hope that these revisions are sufficient to make our manuscript acceptable for publication in Nanomaterials. If you believe that any additional clarifications need to be addressed, I will be happy to include them. Once again, thank you very much for your comments and suggestions.

Yours sincerely,
Cheng-Te Lin
Ningbo Institute of Materials Technology and Engineering, Chinese Academy of Sciences
E-mail: [email protected]

Round 2

Reviewer 2 Report

In my opinion, after improving the paper is ready to be published

Author Response

Thank you!

Reviewer 3 Report

I still have a few key comments and questions. 

1. There are many prior work on graphite paper which has dramatically higher thermal conductivity which are not included in Table 1 such as the Ref 4 of the manuscript which shows higher cross-plane thermal conductivity up to 60 W/m-K which is greatly higher than this work. In other papers, like "Thermally Conductive Phase Change Composites by Large-Size Oriented Graphite Sheets for Scalable Thermal Energy Harvesting. Adv. Mater. 2019, 31 (49), 201905099", the thermal conductivity can be > 120 W/m-K and the sample can be easily cut and reoriented such that the graphene basal planes are vertical to the heat source surface for heat dissipation application. I still highly doubt the novelty of this work. There is a huge number of prior publications on thermal conductivity enhancement in graphene composite such that the data in Table 1 is far from complete. More comprehensive comparison with prior work is necessary. 

2. How is Cp measured in Table 2?

3. The Cp values for the simulation domains makes no sense in Table 3. These are not the actual Cp of the materials. 

4. What is φ in Table 3 and the text below? 

Author Response

Dear editor,

We are very grateful to your and the reviewers’ critical comments and thoughtful suggestions. Based on these comments and suggestions, we have made careful revision on the original manuscript entitled " A Hierarchically Structured Graphene/Ag Nanowires Paper As Thermal Interface Materials " (Manuscript ID: nanomaterials-2163052). We responded point by point to the comments of reviewer #3 as listed below, along with a clear indication of the location of the revision.

In specific response to the points raised by reviewer #3:

Question 1: There are many prior work on graphite paper which has dramatically higher thermal conductivity which are not included in Table 1 such as the Ref 4 of the manuscript which shows higher cross-plane thermal conductivity up to 60 W/m-K which is greatly higher than this work. In other papers, like "Thermally Conductive Phase Change Composites by Large-Size Oriented Graphite Sheets for Scalable Thermal Energy Harvesting. Adv. Mater. 2019, 31 (49), 201905099", the thermal conductivity can be > 120 W/m-K and the sample can be easily cut and reoriented such that the graphene basal planes are vertical to the heat source surface for heat dissipation application. I still highly doubt the novelty of this work. There is a huge number of prior publications on thermal conductivity enhancement in graphene composite such that the data in Table 1 is far from complete. More comprehensive comparison with prior work is necessary.

Reply: Thank you for the comment. As described by the reviewer, the graphene papers or graphene composites are cut and reoriented as TIMs. TIMs prepared through this method have a high through-plane thermal conductivity. However, these TIMs have some problems, such as complex preparation process, thick thickness and large compression modulus, which limit their use in practical applications. Like “High-performance thermal interface materials consisting of vertically aligned graphene film and polymer. Carbon 2016,(109), 552-557”, the compression modulus can be 6.506 MPa and Young's modulus can be >500 MPa. Like “Rational design of high-performance thermal interface materials based on gold-nanocap-modified vertically aligned graphene architecture”, the compression modulus is not significantly reduced via adding porous polymer and the thickness of sample is always >0.5 mm. Fluffy graphene papers can be prepared quickly, but ultra-low through-plane thermal conductivity limits them use as TIMs. Constructing additional heat flow pathways along the through-plane direction inside the graphene papers by inserting the thermally conductive materials between horizontal graphene layers also attracts researchers' attention. In this work, the method can quickly prepare TIMs with large-size (diameter >280 mm), ultra-thin (thickness <0.1 mm) and low compression modulus (< 16.8 kPa).

Question 2: How is Cp measured in Table 2?

Reply: Thank you for the comment. Figure (a) show the heat flow curves of GP, IGAP and the sapphire obtained by differential scanning calorimetry (DSC). Figure (b) show the difference of heat flow between measured samples and baseline per unit mass ( ΔH/m ) for calculating specific heat capacity based on the equation:

×

Figure (c) shows the Cp of the samples from 25 ℃ to 150 ℃.

Question 3: The Cp values for the simulation domains makes no sense in Table 3. These are not the actual Cp of the materials.

Reply: Thank you for the comment. The authors apologize for the mistake. Based on your suggestion, the authors changed the actual Cp of the materials in the heat analysis. The corresponding description also have been added in the revised manuscript. (Table 3, line 234)

Question 4: What is φ in Table 3 and the text below? 

Reply: Thank you for the comment. The authors apologize for the mistake and corrected φ to Φ on the revised manuscript. The corresponding description also have been added in the revised manuscript. (line 241 and Table 3)

We appreciate for your warm work earnestly, and hope that the correction will meet with approval. The manuscript has been overall checked, and the changes marked in red. We hope that these revisions are sufficient to make our manuscript acceptable for publication in Nanomaterials. If you believe that any additional clarifications need to be addressed, I will be happy to include them. Once again, thank you very much for your comments and suggestions.

Yours sincerely,
Cheng-Te Lin
Ningbo Institute of Materials Technology and Engineering, Chinese Academy of Sciences
E-mail: [email protected]

Reviewer 5 Report

The present manuscript is properly revised. It meets the criteria for publication.

Author Response

Thank you!
